# Systematic Characterization and Regulatory Role of lncRNAs in Asian Honey Bees Responding to Microsporidian Infestation

**DOI:** 10.3390/ijms24065886

**Published:** 2023-03-20

**Authors:** Zixin Wang, Siyi Wang, Xiaoxue Fan, Kaiyao Zhang, Jiaxin Zhang, Haodong Zhao, Xuze Gao, Yiqiong Zhang, Sijia Guo, Dingding Zhou, Qiming Li, Zhihao Na, Dafu Chen, Rui Guo

**Affiliations:** 1College of Animal Sciences (College of Bee Science), Fujian Agriculture and Forestry University, Fuzhou 350002, China; 2Apitherapy Research Institute of Fujian Province, Fuzhou 350002, China

**Keywords:** lncRNA, ceRNA, honey bee, *Apis cerana cerana*, *Nosema ceranae*, immune response

## Abstract

Long noncoding RNAs (lncRNAs) are pivotal regulators in gene expression and diverse biological processes, such as immune defense and host–pathogen interactions. However, little is known about the roles of lncRNAs in the response of the Asian honey bee (*Apis cerana*) to microsporidian infestation. Based on our previously obtained high-quality transcriptome datasets from the midgut tissues of *Apis cerana cerana* workers at 7 days post inoculation (dpi) and 10 dpi with *Nosema ceranae* (AcT7 and AcT10 groups) and the corresponding un-inoculated midgut tissues (AcCK7 and AcCK10 groups), the transcriptome-wide identification and structural characterization of lncRNAs were conducted, and the differential expression pattern of lncRNAs was then analyzed, followed by investigation of the regulatory roles of differentially expressed lncRNAs (DElncRNAs) in host response. Here, 2365, 2322, 2487, and 1986 lncRNAs were, respectively, identified in the AcCK7, AcT7, AcCK7, and AcT10 groups. After removing redundant ones, a total of 3496 *A. c. cerana* lncRNAs were identified, which shared similar structural characteristics with those discovered in other animals and plants, such as shorter exons and introns than mRNAs. Additionally, 79 and 73 DElncRNAs were screened from the workers’ midguts at 7 dpi and 10 dpi, respectively, indicating the alteration of the overall expression pattern of lncRNAs in host midguts after *N. ceranae* infestation. These DElncRNAs could, respectively, regulate 87 and 73 upstream and downstream genes, involving a suite of functional terms and pathways, such as metabolic process and Hippo signaling pathway. Additionally, 235 and 209 genes co-expressed with DElncRNAs were found to enrich in 29 and 27 terms, as well as 112 and 123 pathways, such as ABC transporters and the cAMP signaling pathway. Further, it was detected that 79 (73) DElncRNAs in the host midguts at 7 (10) dpi could target 321 (313) DEmiRNAs and further target 3631 (3130) DEmRNAs. TCONS_00024312 and XR_001765805.1 were potential precursors for ame-miR-315 and ame-miR-927, while TCONS_00006120 was the putative precursor for both ame-miR-87-1 and ame-miR-87-2. These results together suggested that DElncRNAs are likely to play regulatory roles in the host response to *N. ceranae* infestation through the regulation of neighboring genes via a *cis*-acting effect, modulation of co-expressed mRNAs via *trans*-acting effect, and control of downstream target genes’ expression via competing endogenous RNA networks. Our findings provide a basis for disclosing the mechanism underlying DElncRNA-mediated host *N. ceranae* response and a new perspective into the interaction between *A. c. cerana* and *N. ceranae*.

## 1. Introduction

Long non-coding RNAs (lncRNAs) are a class of linear RNA molecules transcribed by RNA polymerase II with a length of more than 200 nt [1]. Accumulating studies are indicative of the participation of lncRNAs in an array of physiological and pathological processes of great importance, such as cell proliferation and differentiation, material and energy metabolism, and immune defense, as well as the occurrence and progression of disease [2,3,4,5]. LncRNAs were earlier thought to be the byproducts of the transcription process without protein-coding ability; however, with the development of related knowledge and techniques, lncRNAs were suggested to exert pivotal biological functions in diverse manners, including the direct regulation of neighboring genes’ transcription, indirect modulation of downstream gene expression via the competing endogenous RNA (ceRNA) network, and translation into proteins or small peptides [6]. LncRNAs have been studied in depth in a few model organisms, such as humans and *Arabidopsis thaliana*; however, progress associated with lncRNAs in insects, including honey bees, is limited. Wu et al. [7] conducted a transcriptomic investigation of lncRNAs in *Bombyx mori* and found that a subseries of lncRNAs were potentially engaged in the regulation of biosynthesis, transport, and secretion of silk proteins by serving as ceRNAs or miRNA precursors. On the basis of a transcriptome-wide analysis of lncRNAs, miRNAs, and mRNAs in *Apis mellifera* queens during different oviposition periods, Chen et al. [8] revealed that these ncRNAs may be involved in ovary activation and oviposition processes through the modulation of related genes’ expression. Previously, following the next-generation sequencing of the *Apis mellifera ligustica* workers’ midguts tissues, we identified a total of 6353 lncRNAs and observed that their structural features were analogous to those discovered in other animals and plants [9]. An increasing number of studies have demonstrated that lncRNAs are critical regulators in insect–pathogen interactions; for example Jayakodi et al. [10] who identified 2470 and 1514 long intergenic noncoding RNAs (lincRNAs) in *Apis cerana* and *A. mellifera* based on deep sequencing and bioinformatics, detected that lincRNAs exhibited a tissue-specific expression pattern among seven organs or tissues, and they further discovered that 10 host lincRNAs were induced to activation upon SBV and DWV infection. Our group previously analyzed the expression profile of lncRNAs in the *A. m. ligustica* workers’ midguts challenged by *Nosema ceranae*, and detected that differentially expressed lncRNAs (DElncRNAs) were likely to participate in modulating the host response via various methods, such as a *cis*-acting effect, *trans*-acting effect, miRNA precursors, and ceRNA networks [9]. In addition, we deciphered the expression pattern of lnRNAs in *Apis cerana cerana* 6-day-old larval gut responding to *A. apis* invasion, and investigated the putative regulatory role of DElncRNAs in the host response [11].

*A. cerana* is widely reared in China and many other Asian countries, and plays a crucial role in pollination for considerable crops and wild flowers, as well as production of api-products [12]. *A. cerana* is the original host for *N. ceranae*, a widespread fungal parasite causing bee nosemosis, which results in severe losses for the apicultural industry [13]. *A. c. cerana*, the nominate subspecies of *A. cerana*, is a major bee species used in beekeeping practices in China. We previously investigated the dynamics of miRNAs in the *A. c. cerana* workers’ midguts infected by *N. ceranae* and revealed the potential role of DEmiRNAs in host immune responses [14]. Recently, we analyzed the differential expression pattern of circular RNAs (circRNAs) in the midguts of *A. c. ceran*a workers during *N. ceranae* infection, and we uncovered that DEcircRNAs were likely to modulate host cellular and humoral immune responses through the regulation of the transcription of source genes or absorption of target miRNAs to affect the downstream gene expression [15]. As a vital layer of post-transcription regulation, lncRNAs are able to not only directly modulate gene expression, but also interact with other ncRNAs such as miRNAs to indirectly regulate the expression of target genes [16]. However, compared with *A. mellifera*, studies on lncRNAs in *A. cerana* are scarce at present. Whether and how lncRNAs participate in the interaction between *A. cerana* and *N. ceranae* has been unclear until now.

Our team previously conducted deep sequencing of cDNA libraries from the midgut tissues of *A. c. cerana* workers at 7 days post inoculation (dpi) and 10 dpi (AcT7 and AcT10 groups), as well as corresponding uninoculated workers’ midgut tissues (AcCK7 and AcCK10 groups) [9]. In this current work, on the basis of the obtained high-quality transcriptome datasets, the systematic identification and structural characterization of lncRNAs in the *A. c. cerana* workers’ midguts were conducted, and the differential expression profile of lncRNAs during the *N. ceranae* infestation was then analyzed in depth, followed by further investigation of the putative regulatory functions of DElncRNAs in host response, with a focus on immune response. Our findings can offer not only a foundation for identifying the mechanism underlying the DElncRNA-regulated response to *N. ceranae* infestation, but also a novel insight into the Asian honey bee microsporidian interaction.

## 2. Results

### 2.1. Quantity and Structural Properties of A. c. cerana lncRNAs

Here, 2365, 2322, 2487, and 1986 lncRNAs were, respectively, identified in the AcCK7, AcT7, AcCK10, and AcT10 groups; the numbers of known lncRNAs were 1001, 960, 1089, and 799, while those of novel lncRNAs were 1364, 1362, 1398, and 1187, respectively. Among the identified lncRNAs, the three most highly expressed were XR_001765028.1, XR_001765880.1, and XR_001766991.1. Additionally, it was found that 1558 lncRNAs were shared by the abovementioned four groups, whereas the quantities of unique ones were 131, 132, 177, and 70, respectively (Figure 1A). After removing redundant ones, a total of 3496 *A. c. cerana* lncRNAs (Appendix A) were ultimately identified, with a length distribution range of 188 nt~441,800 nt (Figure 1B). As shown in Figure 1C, the length of introns in lncRNAs was smaller than that in mRNAs: 1254 (35.90%) lncRNAs contained only one intron, while the remaining 2242 (64.10%) included two or more introns (Figure 1D). Similarly, the length of exons in lncRNAs was shorter than that in mRNAs (Figure 1E); as many as 2162 (61.84%) lncRNAs contained two exons, and another 1334 (38.16%) lncRNAs included more than two exons (Figure 1F).

### 2.2. Differential Expression Profile of lncRNAs in the Midguts of A. c. cerana Workers Responding to N. ceranae Infestation

In the AcCK7 vs. AcT7 comparison group, 50 upregulated and 29 downregulated lncRNAs were identified (Figure 2A, see also Appendix A); among these, XR_001767038.1 (log_2_Fold change = 12.47) (log_2_FC = 12.47), TCONS_00047233 (log_2_FC = 10.55), and XR_001766127.1 (log_2_FC = 10.44) were the three most upregulated lncRNAs, while TCONS_00006288 (log_2_FC = −10.76) was the most downregulated one, followed by XR_001765791.1 (log_2_FC = −10.59) and XR_001766607.1 (log_2_FC = −10.37) (Figure 2B). Comparatively, 26 upregulated and 47 downregulated lncRNAs were detected in the AcCK2 vs. AcT2 comparison group (Figure 2C, see also Appendix A); among these, three lncRNAs including TCONS_00029732 (log_2_FC = 13.07), TCONS_00018981 (log_2_FC = 12.45), and XR_001767090.1 (log_2_FC = 11.93) were the most upregulated, whereas TCONS_00004222 (log_2_FC = −15.01) was the most downregulated one, followed by TCONS_00023581 (log_2_FC = −11.55) and XR_001766877.1 (log_2_FC = −11.46) (Figure 2D). Further analysis showed that there were six shared DElncRNAs (TCONS_00011721, TCONS_00022015, TCONS_00024562, XR_001765500.1, XR_001765515.1, and XR_001766127.1) by the aforementioned two comparison groups, while the numbers of unique ones were 72 and 67, respectively (Figure 2E). Detailed information regarding DElncRNAs in AcCK7 vs. AcT7 and AcCK7 vs. AcT10 comparison groups are, respectively, presented in Appendix A.

### 2.3. Analysis of the Cis-Acting Effect of DElncRNAs on Host Response to N. ceranae Infestation

Accumulating evidence showed that lncRNAs are capable of regulating the expression of one or several linked and continuous genes in close genomic proximity (called the *cis*-acting effect) [17]. It was suggested that 79 DElncRNAs in the AcCK7 vs. AcT7 in the comparison group potentially regulated 87 up- and downstream genes, which were engaged in 30 Gene Ontology (GO) terms relevant to biological process, cellular component, and molecular function, such as metabolic process, cell part, and transporter activity (Figure 3A, see also Appendix A). In contrast, 73 DElncRNAs in the AcCK7 vs. AcT10 comparison group putatively modulated 87 up- and downstream genes, which were enriched in 13 functional terms such as regulation of biological process, muti-organism process, and binding (Figure 3B, see also Appendix A).

In addition, Kyoto Encyclopedia of Genes and Genomes (KEGG) pathway analysis demonstrated that the up- and downstream genes in the workers’ midguts at 7 dpi were associated with 82 pathways, including apoptosis, adhesion junction, and phagosome (Figure 4A, see also Appendix A). Comparatively, the up- and downstream genes in the workers’ midguts at 10 dpi were relative to 29 pathways, such as purine metabolism, RNA degradation, and endocytosis (Figure 4B, see also Appendix A).

### 2.4. Investigation of the Trans-Acting Effect of DElncRNAs on the Host Response to N. ceranae Infestation

In addition, those lncRNAs diffusing from the synthesis site were able to act directly on many genes at a great distance at the same chromosome or other chromosomes (named *trans*-acting effect) [18]. It was detected that a total of 235 mRNAs were co-expressed with DElncRNAs in the AcCK7 vs. AcT7 comparison group (Appendix A); among these, XR_001765926.1 was found to co-express with the largest number of mRNAs (90). For example, the co-expressed zinc transporter ZIP11 protein encoding gene (XM_017059863.1) had a positive correlation with XR_001766833.1, but had a negative correlation with TCONS_00021820; the co-expressed SEC23-like interacting protein isoform X2 encoding gene (XM_017063532.1) was positively correlated with XR_001766919.1, while it was negatively correlated with XR_001765791.1. Meanwhile, 209 mRNAs were observed to be co-expressed with DElncRNAs in the AcCK10 vs. AcT10 comparison group (Appendix A); among these, TCONS_00029732, XR_001765691.1, and XR_001766718.1 were observed to co-express with the largest number of mRNAs (88) (Appendix A). For instance, the co-expressed DNA mismatch repair protein MSH2 encoding gene (XM_017051143.1) had a positive correlation with TCONS_00007982 and XR_001766995.1; the co-expressed alkaline phosphatase 4 protein-encoding gene (XM_017048382.1) was negatively correlated with TCONS_00047837, TCONS_00004245, and TCONS_00044926 (Appendix A); and the co-expressed promoter complex subunit 7 (XM_017063362.1) was negatively correlated with XR_001765313.1 and TCONS_00004222 (Appendix A).

### 2.5. Analysis of DElncRNA-Involved ceRNA Regulatory Networks

Various RNA molecules, including lncRNAs, can competitively bind to miRNAs to modulate the expression of downstream genes and further regulate biological processes such as host response and pathogen infection (called ceRNA mechanism) [19]. Here, nine DElncRNAs in the workers’ midguts at 7 dpi were observed to potentially target eight DEmiRNAs, and further bound to twenty-five DEmRNAs, forming five independent subnetworks (Figure 5A); additionally, TCONS_00022015 targeted both miR-1-x and miR-598-y, TCONS_00022015 targeted both miR-252-y and miR-6717-x, while the other seven DElincRNAs had only one target (Figure 5A). Comparatively, six DElncRNAs in the workers’ midguts at 10 dpi were found to target five DEmiRNAs and further link to eight DEmRNAs, forming five independent subnetworks (Figure 5B); additionally, TCONS_00007982 and XR_001765691.1 both targeted novel_m0003-3p, while the other four DElncRNAs had only one target. (Figure 5B) Detailed information on the targeting relationships among DElncRNAs, DEmiRNAs, and DEmRNAs in the AcCK7 vs. AcT7 and AcCK10 vs. AcT10 comparison groups are shown in Appendix A.

### 2.6. Analysis of DElncRNAs as Putative miRNA Precursors

As a kind of versatile regulator, certain lncRNAs were verified to function as precursor molecules that are processed into miRNAs, greatly enlarging the reservoir of miRNAs within cells [20]. In the workers’ midguts at 7 dpi, TCONS_00024312 and XR_001765805.1 were predicted to be precursors for ame-miR-315 and ame-miR-927, respectively (Figure 6A). As shown in Figure 6B, TCONS_00006120 in the workers’ midguts at 10 dpi was found to be a putative miRNA precursor targeting the mature sequences of both ame-miR-87-1 and ame-miR-87-2.

### 2.7. Validation of DElncRNAs by RT-qPCR

The primers for RT-qPCR assay were subjected to PCR amplification and agarose gel electrophoresis; the results showed that the expected fragments were amplified using each primer pair (Appendix A), verifying the specificity of these primers. The RT-qPCR results indicated that the expression trends of the randomly selected 12 DElncRNAs were consistent with those in the sequencing data (Figure 7), indicative of the authenticity and reliability of the transcriptome datasets used in this work.

## 3. Discussion

### 3.1. Quantity, Structural Property, and Expression Pattern of lncRNAs in the Midguts of A. c. cerana Workers Infected by N. ceranae

Here, a total of 3031 lncRNAs with a length distribution ranging from 188 nt to 441,800 nt were for the first time identified in *A. c. cerana* midguts. In view of the limited quantity of lncRNAs in *A. cerana*, the identified lncRNAs could offer valuable resources for the study of lncRNAs in *A. c. cerana* and other subspecies belonging to *A. cerana*. Given that the expression of lncRNAs is the tissue, developmental, and stress stages, the identified lncRNAs were speculated to be only a fraction of the total lncRNAs in *A. c. cerana*. Additionally, the identified *A. c. cerana* lncRNAs were detected to have shorter lengths of exons and introns, analogous to the characteristics of lncRNAs discovered in other species [21,22].

An array of studies showed that pathogen infections affected the expression pattern of lncRNAs in animals and plants [23,24,25,26]. In this work, 79 and 73 DElncRNAs were screened in the midguts of *A. c. cerana* workers at 7 dpi and 10 dpi with *N. ceranae*, respectively, indicative of the overall change of host lncRNAs caused by *N. ceranae* infestation. Further analysis suggested that six DElncRNAs, such as TCONS_00011721, TCONS_00022015, TCONS_00024562, XR_001765500.1, XR_001765515.1, and XR_001766127.1, are shared by the abovementioned comparison groups, implying that these common DElncRNAs may be vital regulators in host response to *N. ceranae* infestation, thus deserving further investigation.

### 3.2. DElncRNAs Were Potentially Involved in Modulating Host Response to N. ceranae Infestation via Both cis- and Trans-Acting Effects

The action modes of lncRNAs are flexible and diverse [3]. One major regulatory manner of lncRNAs is the *cis*-acting effect, which has been verified in numerous animals, such as *Plutella xylostella* [27] and *A. m. ligustica* [9]. LncRNAs located upstream and downstream of the protein-encoding genes may generate partial overlap with promoters, enhancers, or inducible elements, thus exerting regulatory functions in the expression of neighboring protein-coding genes at the transcriptional or post-transcriptional level [2,3]. The Notch signaling pathway is highly conserved in various insects and plays critical roles in controlling organ development and tissue homeostasis [28,29]. The Mitogen-activated protein kinases (MAPK) signaling pathway acts as a signaling hub that can be activated by external signals, such as mitogens, hormones, temperature, and osmolality [30]. Additionally, a series of signaling pathways, including MAPK, JNK, ERK, P53, and PI3K/Akt, jointly regulate diverse insect biological processes, such as eye development, cell shape, cellular apoptosis, neural development, embryonic dorsal plate closure, and immune response [28,31]. The Notch and MAPK signaling pathways could synergistically modulate insect growth, development, and immunity [32,33]. Here, we observed that two neighboring genes of DElncRNAs in the host midguts at 7 dpi, such as the Nicastrin protein isoform X2 encoding gene (*NCSTN*) (XM_017058633.1) and γ-secretase protein subunit Aph-1 encoding gene (*APH-1*) (XM_017060355.1)), were upregulated and enriched in the Notch signaling pathway; the vein-like protein encoding gene *Vein* (XM_017059352.1) in the host midguts at 10 dpi was upregulated and enriched in the MAPK signaling pathway. These results indicate that the corresponding host DElncRNAs (TCONS_00032459, XR_001766455.1, and TCONS_00034390) were putatively engaged in regulating the Notch and MAPK signaling pathways in a *cis*-acting manner during *N. ceranae* infestation (Figure 8).

Insects lack adaptive immunity and rely on innate immunity in defense against pathogen invasion [32]. Insect innate immunity includes cellular immune response, such as phagocytosis, colocalization, encapsulation reaction, melanization, ubiquitin-mediated protein hydrolysis, and lysosome, as well as humoral immune responses, represented by the MAPK, Jak-STAT, and NF-κB signaling pathways [28,34,35,36,37,38,39,40]. Here, three, two, one, and one neighboring genes of DElncRNAs relevant to phagosome, endocytosis, autophagy, and lysosome, respectively, were detected in the workers’ midguts at 7 dpi, indicating that the corresponding DElncRNAs were potentially involved in the *N. ceranae* response of the host by regulating the aforementioned four vital cellular immune pathways with a *cis*-acting effect.

P53, a tumor suppressor protein that acts as a transcription factor, affected the release of apoptosis-forming factors, such as cytochrome c in mitochondria, by downregulating *Bcl-2* expression and, meanwhile, upregulating *Bax* expression, which, in turn, controlled the cell cycle [41]. The p53 signaling pathway was involved in maintaining genomic stability and regulating DNA damage repair, as well as apoptosis [41]. Here, we observed the upregulation of the cytochrome c protein encoding gene (*CytC*) (XM_017063771.1), enriched in both apoptosis and the p53 signaling pathway; the universal transcription factor IIH subunit 3 encoding gene (*TTDA*) (XM_017063798.1), enriched in nucleotide excision repair, was also detected as upregulated. Together, these results demonstrate that corresponding DElncRNAs may be engaged in regulating apoptosis, the p53 signaling pathway, and nucleotide excision repair through activating the expression of *CytC* and *TTDA* via a *cis*-acting manner, further affecting the host response to *N. ceranae* infestation. However, additional work is required to dissect the underlying mechanisms of the DElncRNA-regulated *cis*-acting manner. LncRNAs are capable of modulating gene expression via interaction with their co-expressed mRNAs [42,43]. In the present study, co-expressed genes with DElncRNAs in the AcCK7 vs. AcT7 and AcCK10 vs. AcT10 comparison groups were associated with a suite of functional terms, such as stress response, catalytic activity, and biological regulation; six cellular and humoral immune pathways, such as ubiquitin-mediated protein hydrolysis, endocytosis, lysosome, autophagy, and MAPK signaling pathways; 11 signaling pathways associated with growth and development including Wnt, FoxO, Hippo, Hedgehog, mTOR, and p53; and 42 pathways relative to material and energy metabolisms, such as carbon metabolism, sulfur metabolism, and oxidative phosphorylation [9,28,34,35,36,37,38,39,40] (Appendix A). The results showed that DElncRNAs may affect many aspects of host midguts’ growth and development, metabolism, and immune defense in a *trans*-acting manner, further responding to *N. ceranae* infestation. More efforts are needed to further ascertain the mechanisms underlying DElncRNA-mediated *trans*-acting effects in the future.

### 3.3. DElncRNAs May Play a Role in Host Response to N. ceranae Infestation by Serving as miRNA Precursors

MiRNAs are key regulators in gene expression and considerable biological processes, such as immune response and apoptosis [44,45]. Increasing studies indicate that lncRNAs can act as miRNA precursors to generate abundant mature miRNAs through a series of splicing and catalysis [14]. By using the immunohistochemistry and luciferase reporter system, Yuan et al. [44] reported that miR-315 affected the structure and function of *Drosophila* neurosynapses by repressing the expression of the target gene *dFMR1*, followed by the knockdown of miR-315, which led to embryonic death, while the overexpression of miR-315 gave rise to pupation defects. Additionally, it is suggested that miR-315 is mainly expressed in the nervous system of *Drosophila* and acts as a specific activator of the wingless signaling pathway, which is involved in regulating wing development [45]. MiR-927 was observed to affect the expression of antimicrobial peptides associated with the Toll signaling pathway of *Aedes albopictus* through the inhibition of serine and to promote DENV infection in humans [46]. He et al. [47] detected that overexpression of miR-927 in *Drosophila melanogaster* liposomes resulted in the downregulation of the expression of the target gene *krl-h1*, which resulted in a significant decrease in egg production, increased embryonic lethality, and pupation defects. MiR-87, a key regulator of *Drosophila* neuronal dendrites C4da, was found to promote neuronal dendrite regeneration by targeting the transcriptional repressor *Ttk69* [48]. Here, three DElncRNAs (TCONS_00024312, XR_001765805.1, and TCONS_00006120) were observed to be potential precursors for miR-315, miR-927, and miR-87. It is speculated that these three DElncRNAs may participate in the modulation of gut development and AMP synthesis by generating abundant miRNAs, further regulating host *N. ceranae* response.

### 3.4. DElncRNAs May Play a Part in Host Response to N. ceranae Infestation through ceRNA Regulatory Networks

MRE-containing lncRNAs can absorb miRNAs via ceRNA networks, thereby reducing the binding of miRNAs to target mRNAs and attenuating the repressive effect on target genes [49,50]. Liu et al. [51] discovered that ame-miR-279a was highly expressed in Kenyon cells in the mushroom body of the brains of *A. mellifera* nurse and forager bees, and the overexpression of ame-miR-279a resulted in an enhanced proboscis extension response to sucrose solution, whereas the knockdown of ame-miR-279a led to a diminished proboscis extension response. Here, two DElncRNAs (XR_001766050.1 and TCONS_00024562) in the host midguts at 7 dpi and three DElncRNAs (TCONS_00024562, XR_001765691.1 and TCONS_00042438) in the host midguts at 10 dpi were observed to target miR-279-y, suggesting that these five DElncRNAs, serving as “molecular sponges”, may be involved in the *N. ceranae* response of *A. c. cerana* workers.

In *Drosophila*, miR-315 was a vital regulator in synaptic structure and transmission by targeting *dfmr1* [44]. Kitatani et al. documented that miR-87 promoted dendrite regrowth during regeneration, at least in part through suppressing Ttk69 in *Drosophila* sensory neurons, suggesting that developmental- and injury-induced dendrite regeneration share a common intrinsic mechanism to reactivate dendrite growth [48]. Here, ten and four DElncRNAs in the abovementioned two comparison groups were detected to target miR-315-x, and seven and four DElncRNAs putatively targeted miR-87-y. The results showed that corresponding DElncRNAs may be engaged in modulating signal transmission and midgut growth and regeneration of workers’ midguts during *N. ceranae* infestation.

The overexpression of miR-927 in the *Drosophila* larval fat body significantly decreased the expression of *Kr-h1* and resulted in reduced oviposition, increased mortality, delayed pupation, and reduced pupal size [47]. In mosquito cell response to viral infection, both the overexpression and knockdown of miR-927 led to the alteration of the expression levels of antimicrobial peptides involved in the Toll signaling pathway [46]. Here, five and three DElncRNAs in the host midguts at 7 dpi potentially targeted miR-927-x and miR-927-y, respectively. This indicates that these DElncRNAs may be involved in the host immune response to *N. ceranae* infestation.

miR-1, a conserved miRNA family that are abundantly expressed in mammals and insects, are widely involved in growth, development, metabolism, and immunity [52,53]. Tci-miR-1-3p was proved to be closely correlated with the resistance of *Tetranychuscinnabarinus* to cyflumetofen [52]. In *Bactrocera dorsalis*, miRNA-1-3p is an important sex determinant in the early male embryo and suppresses the expression of *Bdtra* [53]. We detected that TCONS_00022015 in the host midguts at 7 dpi putatively targets miR-1-x, which further targets max-binding protein MNT-like transcript variant X3 encoding gene (XM_017051758.1) and uncharacterized protein encoding gene (XM_017062549.1); these targets were enriched in the Hedgehog pathway. We inferred that TCONS_00022015 may be employed by the host to absorb miR-1-x to regulate the Hedgehog signaling pathway and ubiquitin-mediated protein hydrolysis in the *N. ceranae* response of workers’ midguts.

In *Drosophila*, miR-252 exerted regulatory functions in metamorphosis, cell cycle, wing formation, and body weight [54]. *A. albopictus* miR-252 inhibited the replication of DENV by regulating the expression of envelope protein genes [55]. Verma et al. reported that the absence of miR-60 expression in the gut of *Caenorhabditis elegans* resulted in the upregulation of genes relative to lysosomal protease, exogenous metabolism, and immune defense, as well as the activation of endocytosis, thereby promoting adaptation to chronic oxidative stress [56]. miR-60 in nematode intestines was suggested to participate in controlling the response to heat and cold stress by regulating the expression of *zip-10* [57]. Here, we observed that TCONS_00032996 potentially targeted miR-252-y, further targeting the telomerase-binding protein EST1A-like isoform X2 encoding gene (XM_017057307.1), a member closely associated with the mRNA surveillance pathway; TCONS_00032996 putatively targeted miR-252-y, further targeting the enriched telomerase-binding protein EST1A-like isoform X2 encoding gene (XM_017057307.1), enriched as well as the mRNA surveillance pathway. As structure-specific and multifunctional enzymes, flap endonucleases are widely observed in eukaryotes and are involved in DNA replication and repair, thus playing a critical part in maintaining genomic stability. Here, TCONS_00035424 in the host midguts at 10 dpi was found to target miR-60-y, which further targeted the gene encoding the flap endonuclease GEN isoform X1 (XM_017063363.1). In summary, the results demonstrated that TCONS_00032996, TCONS_00032996, and TCONS_00035424 in the workers’ midguts were likely to competitively bind to miR-252 and miR-60 to modulate the mRNA surveillance pathway and DNA replication and repair, further mediating the host response to the infestation by *N. ceranae*. Recently, we established the technical platform for the functional investigation of honey bee lncRNAs and miRNAs [58,59]. In the near future, we will conduct RNAi of these promising candidate DElnRNAs mentioned above to further explore their ceRNA functions in the response of *A. c. cerana* workers’ midguts to *N. ceranae* invasion.

## 4. Materials and Methods

### 4.1. Biological Materials

*A. c. ceranae* colonies were reared in the apiary of the College of Animal Sciences (College of Bee Science), Fujian Agriculture and Forestry University, Fuzhou city, China. *N. ceranae* spores were previously purified [14,60] and conserved at the Honey Bee Protection Laboratory, Fujian Agriculture and Forestry University, Fuzhou city, China.

### 4.2. Transcriptome Data Source

On the basis of our previously established method, the midgut tissues of *A. c. cerana* workers at 7 dpi and 10 dpi with *N. ceranae* spores (AcT7 and AcT10 groups), as well as corresponding uninoculated workers’ midgut tissues (AcCK7 and AcCK10 groups), were, respectively, dissected and then subjected to RNA isolation, cDNA library construction, and Illumina sequencing (Guangzhou Gene Denovo Biotechnology Co., Ltd., Guangzhou, China), followed by strict quality control of the raw data [15], which were deposited in the NCBI SRA database (https://www.ncbi.nlm.nih.gov/sra (accessed on 5 September 2022)) and linked to BioProject number: PRJNA562787 [14].

### 4.3. Prediction and Characterization of lncRNAs

Following the method described by Chen et al. [9], a combination of CPC [61] and CNCI [62] software was used to predict lncRNAs in the AcCK7, AcCK10, AcT7, and AcT10 groups, and the intersection of the prediction results was regarded as reliable lncRNAs. The expression level of each lncRNA was normalized to FPKM (Fragments Per Kilobase Million). The OmicShare platform (https://www.omicshare.com/, accessed on 5 September 2022) was used to perform a Venn analysis of lncRNAs in the aforementioned four groups. Next, the transcript length, intron length, intron number, exon length, and exon number of the lncRNAs and mRNAs were counted and compared, followed by visualization using GraphPad Prism 6.0 software (San Diego, CA, USA).

### 4.4. Analysis of DElncRNAs

The DElncRNAs in the AcCK7 vs. AcT7 and AcCK10 vs. AcT10 comparison groups were screened using edgeR software (version 4.2) [63] (https://bioconductor.org/packages/release/bioc/html/edgeR.html/, accessed on 5 September 2022), with the criteria of |log_2_FC| > 1 and *p* < 0.05. The Venn analysis and expression clustering of DElncRNAs were performed using a related tool in the OmicsShare platform (https://www.omicshare.com/, accessed on 5 September 2022)), with default parameters.

### 4.5. Investigation of the Cis-Acting and Trans-Acting Effects of DElncRNAs

According to the method described by Ye et al. [64], neighboring genes within 10 kb upstream and downstream of DElncRNAs were surveyed and considered as potential targets. Subsequently, using the Blast tool in the NCBI database (https://www.ncbi.nlm.nih.gov/, accessed on 5 September 2022), the neighboring genes were, respectively, aligned to the GO (http://www.geneontology.org/, accessed on 5 September 2022) and KEGG (http://www.genome.jp/kegg/, accessed on 5 September 2022) databases to the gain corresponding functional and pathway annotations.

The co-expressed genes were predicted by correlation analysis or co-expression analysis of lncRNAs and protein-coding genes in the samples. The Pearson correlation coefficient method was used to analyze the correlation between lncRNAs and protein-coding genes between samples, and only the maximum positive correlation result and the maximum negative correlation result were retained. Next, the interactions between DElncRNAs and mRNAs were visualized using Cytoscape (version 3.7.2) (Bethesda, MD, USA) [65]. Further, the co-expressed mRNAs were subjected to GO term and KEGG pathway analyses utilizing the Blast tool.

### 4.6. Target Prediction and ceRNA Network Analysis

Following the method described by Guo et al. [66], the DElncRNA-targeted DEmiRNAs and DEmiRNA-targeted DEmRNAs were predicted with TargetFinder software (Livermore, CA, USA, https://targetfinder.org/ (accessed on 5 September 2022)) [67] following the criteria of free energy < −10 and *p* < 0.05. Next, the DElncRNA-DEmiRNA and DElncRNA-DEmiRNA-DEmRNA regulatory networks were constructed on the basis of the predicted targeting relationships, followed by visualization utilizing Cytoscape v3.7.2 software (MD, USA) [58] with the default parameters.

### 4.7. Analysis of the DElncRNAs Serving as miRNA Precursors

Based on the sequence information and structural features of the miRNA precursors, the sequences of the DElncRNAs were matched to the miRBase database (http://www.mirbase.org/, accessed on 5 September 2022) to search for potential miRNA precursors, and only those with comparison coverage > 90% were selected. Meanwhile, the miRNAs and their precursors derived from DElncRNAs were identified using the miRPara software (Wuhan, China, http://www.whiov.ac.cn/bioinformatics/mirpara/, accessed on 5 September 2022) [68] with default parameters. Ultimately, the intersection set was regarded as miRNA precursors with high confidence.

### 4.8. RT-qPCR Validation of DElncRNAs

To verify the reliability of the transcriptome data used in this study, four DElncRNAs in the AcCK7 vs. AcT7 comparison group and eight DElncRNAs in the AcCK10 vs. AcT10 comparison group were randomly selected for RT-qPCR validation. Specific primers (Appendix A) for the selected DElncRNAs were designed with DNAMAN software (version 10) (LynnonBiosoft, San Ramon, CA, USA) and then synthesized by Sangon Biotech (Shanghai) Co., Ltd. (Shanghai, China). The total RNA of the midgut samples (*n* = 3) in the AcCK7, AcT7, AcCK10, and AcT10 groups were isolated using RNA extraction kits (Takara, Dalian, China). Next, reverse transcription was performed using Oligo (dT)_23_ primers to obtain the corresponding cDNAs (Vazymes, Nanjing, China), which were then used as templates for qPCR detection. The *actin* gene (GeneBank ID: 107999330) was used as the internal reference. The 20 μL reaction system contained 1 μL of reverse primer (10.0 μmol·L^−1^), 1 μL of cDNA, 10 μL of SYBR Green Dye (Vazymes, Nanjing, China), and 7 μL of sterile water. The reaction was conducted on a QuanStudio Real-Time PCR System (ThemoFisher, Waltham, MA, USA) following the cycling parameters of 95 °C for 1 min, followed by 40 cycles at 95 °C for 15 s, 60 °C for 30 s, and 72 °C for 45 s. The relative expression level of each DElncRNA was calculated using the 2^–∆∆CT^ method [69]. GraphPad Prism 6.0 software was used for the relevant data analysis and plotting.

### 4.9. Statistical Analysis

All statistical analyses were performed using SPSS software (IBM, Armonk, NY, USA) and GraphPad Prism 6.0 software. The data are shown as the mean ± standard deviation (SD). The statistical analysis was conducted with the independent samples using t-test and one-way ANOVA.

## 5. Conclusions

In conclusion, 1795 known lncRNAs and 1701 novel ones were for the first time identified in *A. c. cerana* workers’ midguts. These *A. c. cerana* lncRNAs shared similar structural characteristics to those discovered in other animals and plants, such as shorter exons and introns than mRNAs. The overall expression profile of lncRNAs in host midguts was changed due to the fact of *N. ceranae* infestation. DElncRNAs may play regulatory roles in host *N. ceranae* response by the regulation of neighboring genes via a *cis*-acting effect, modulation of co-expressed mRNAs via a *trans*-acting effect, and control of downstream target gene expression via ceRNA networks.

## Figures and Tables

**Figure 1 ijms-24-05886-f001:**
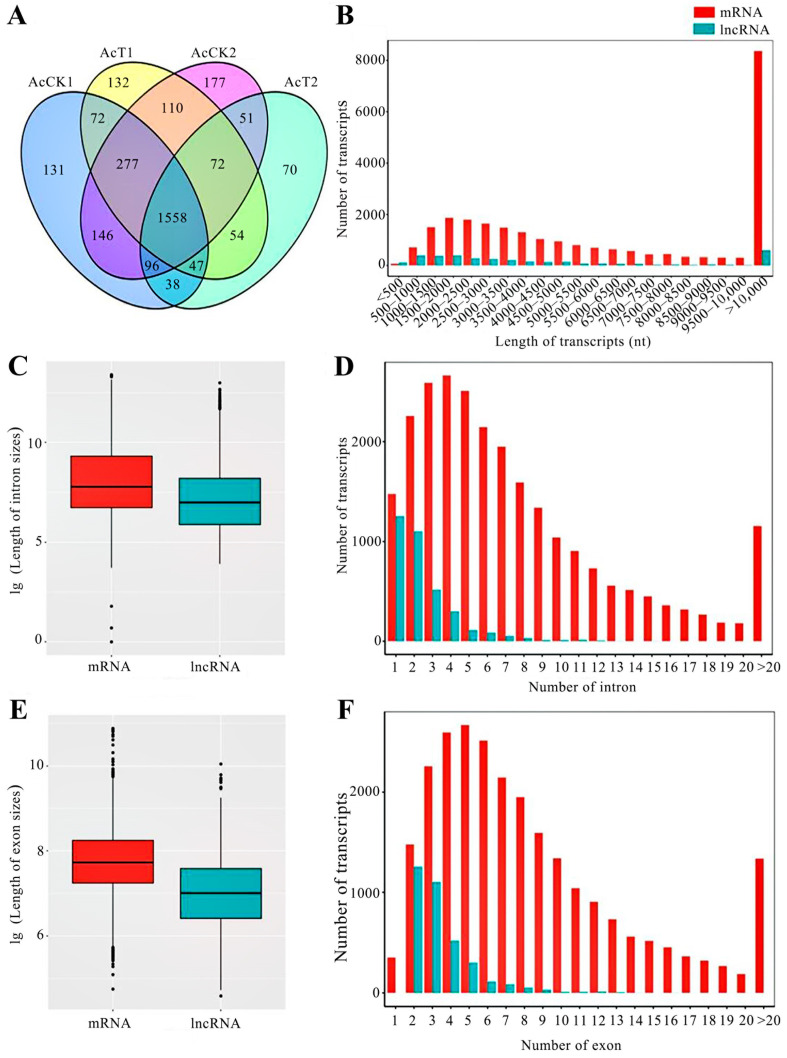
Comparison of the structural characteristics between *A. c. cerana* lncRNAs and mRNAs. (**A**): Venn diagram of the lncRNAs identified in four groups; (**B**): length distribution of lncRNAs and mRNAs; (**C**): overall lengths of introns in lncRNAs and mRNAs; (**D**): number statistics of introns in lncRNAs and mRNAs; (**E**): overall lengths of exons in lncRNAs and mRNAs; (**F**): number statistics of exons in lncRNAs and mRNAs.

**Figure 2 ijms-24-05886-f002:**
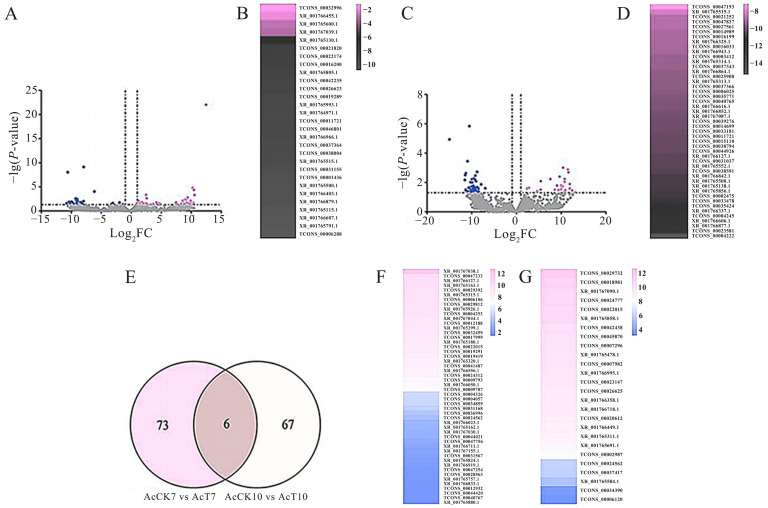
Differential expression profile of lncRNAs involved in response of *A. c. cerana* workers’ midguts to *N. ceranae* infestation. (**A**,**B**): Volcano diagrams of DElncRNAs in the AcCK7 vs. AcT7 and AcCK10 vs. AcT10 comparison groups, DElncRNAs are marked with bule (down), red (up) and grey (not significant), the criteria for a DElncRNAs are |log_2_FC| > 1 and *p* < 0.05; (**C**): Venn diagram of DElncRNAs; (**D**,**E**): expression clustering of up- and downregulated lncRNAs in the AcCK7 vs. AcT7 comparison group; (**F**,**G**): expression clustering of up- and down-regulated DElncRNAs in the AcCK10 vs. AcT10 comparison group.

**Figure 3 ijms-24-05886-f003:**
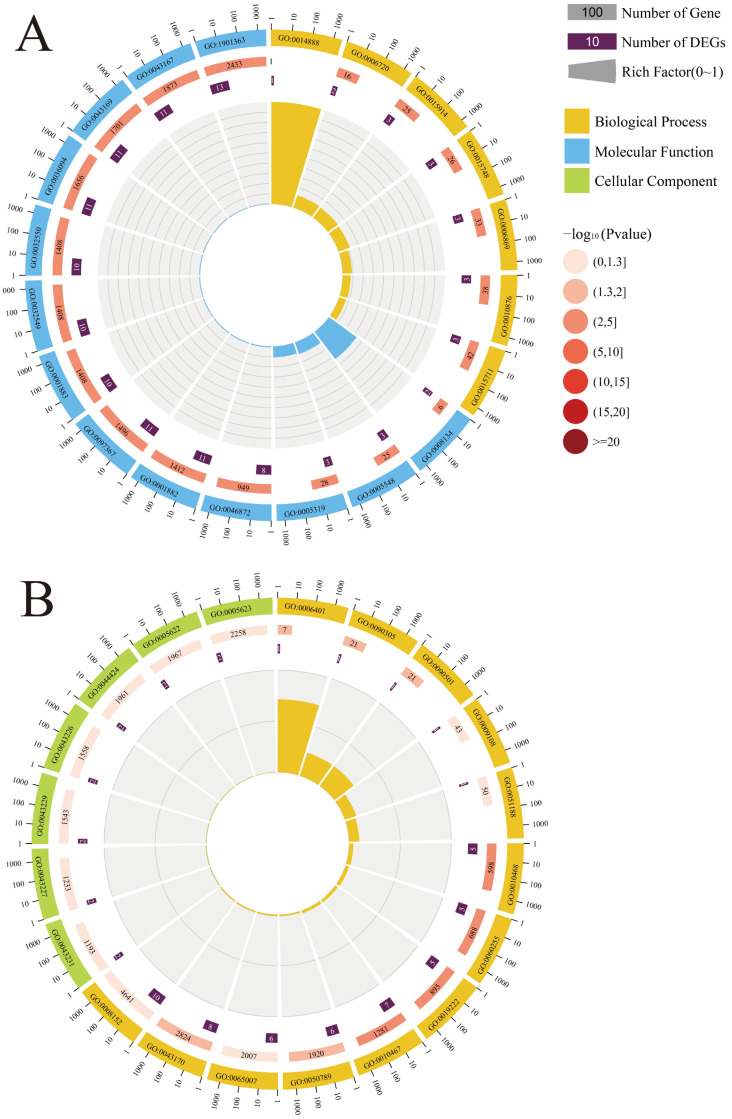
GO classification of the up- and downstream genes of the top 20 DElncRNAs in the AcCK7 vs. AcT7 (**A**) and AcCK7 vs. AcT10 (**B**) comparison groups.

**Figure 4 ijms-24-05886-f004:**
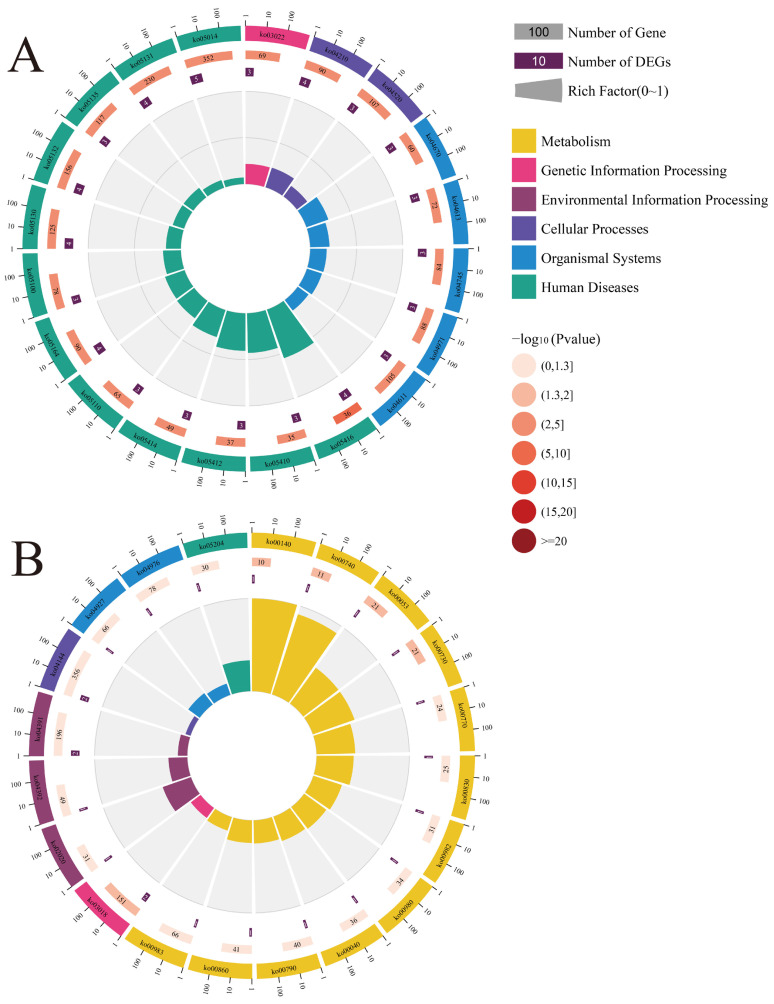
KEGG pathways enriched by the up-and downstream genes of the top 20 DElncRNAs in the AcCK7 vs. AcT7 (**A**) and AcCK10 vs. AcT10 (**B**) comparison groups.

**Figure 5 ijms-24-05886-f005:**
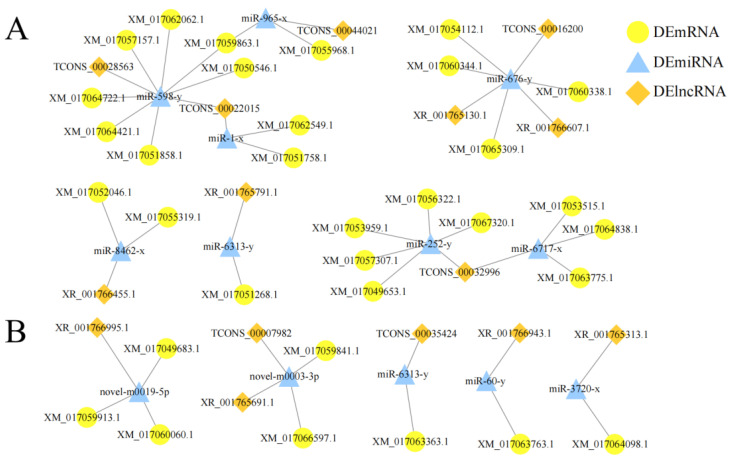
DElncRNA-DEmiRNA-DEmRNA regulatory networks in the AcCK7 vs. AcT7 (**A**) and AcCK10 vs. AcT10 (**B**) comparison groups.

**Figure 6 ijms-24-05886-f006:**
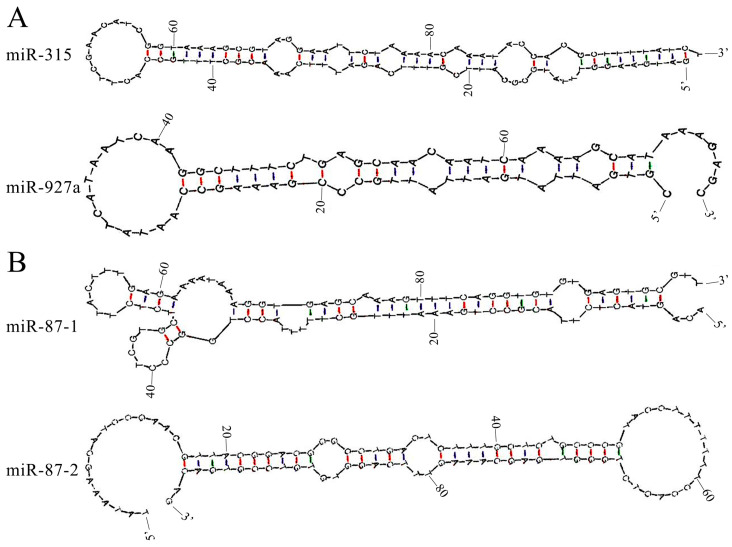
Secondary structures of four miRNA precursors: (**A**) two DElncRNAs as miRNA precursors in the AcCK7 vs. AcT7 comparison group; (**B**) one DElncRNA as miRNA precursor in the AcCK10 vs. AcT10 comparison group.

**Figure 7 ijms-24-05886-f007:**
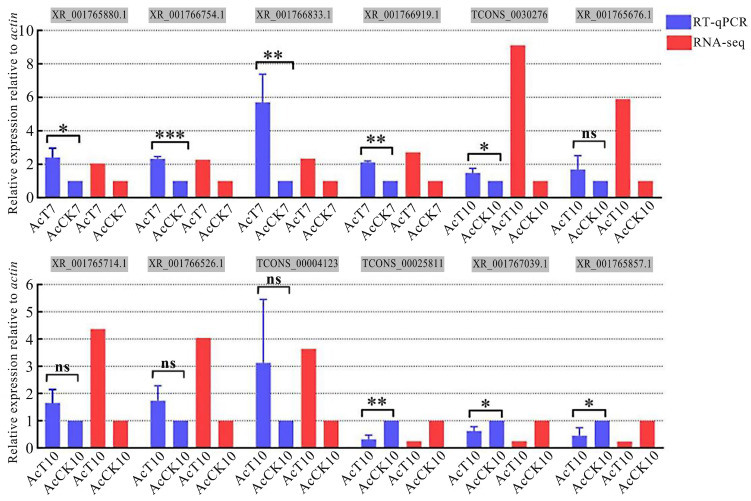
RT-qPCR validation of 12 DElncRNAs. ns, nonsignificant. * *p* < 0.05, ** *p* < 0.01, and *** *p* < 0.001.

**Figure 8 ijms-24-05886-f008:**
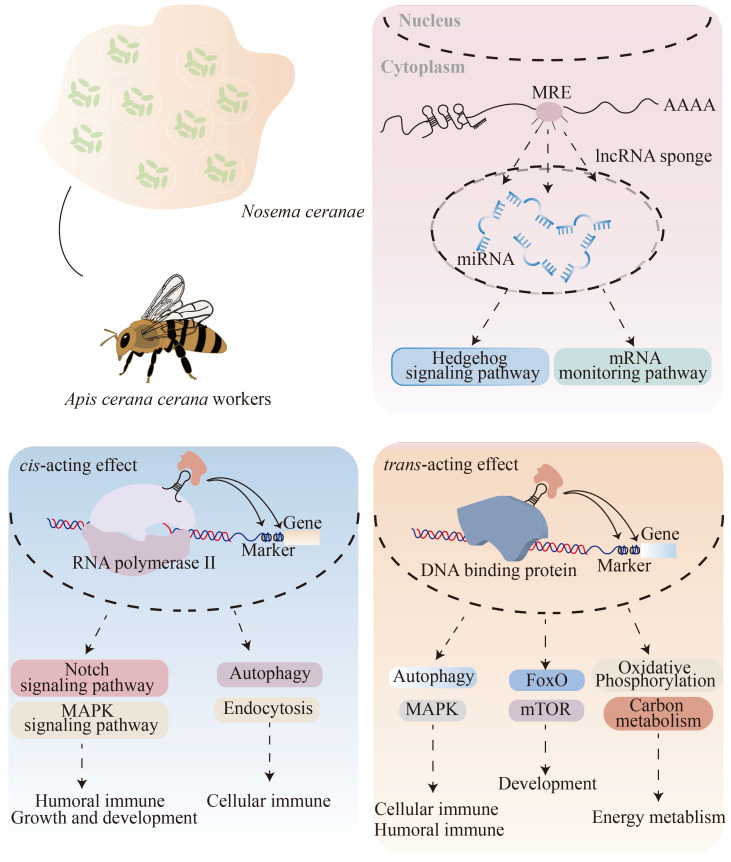
A hypothetical working model of the DElncRNA-modulated immune response of *A. c. cerana* workers’ midguts to *N. ceranae* infestation.

## Data Availability

Raw data generated from RNA-seq are available in the NCBI SRA database under the BioProject number: PRJNA562787 [14].

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
