# Peer review of "Systematic Characterization and Regulatory Role of lncRNAs in Asian Honey Bees Responding to Microsporidian Infestation"

_ijms, 2023, doi:10.3390/ijms24065886_

Round 1
Reviewer 1 Report
The article by Wang et al. entitled “Systematic characterization and regulatory role of lncRNAs in Asian honey bee responding to microsporidian infestation” used transcriptomics to identify long non-coding RNAs, their differential expression patterns in response to microsporidian infestation, and examine the regulatory roles of these differentially expressed long non-coding RNAs in host responses. I have several concerns before this manuscript can be accepted for publication.
Major concerns:
- The lncRNAs identified were said to range in size from 188 bp to 441,800 bp. This does not make sense because lncRNAs, by definition, are larger than 200 bp in length.
- This manuscript requires professional English language editing prior to its potential acceptance.
- There should be a venn diagram in Figure 1 showing how many of the identified lncRNAs were already known to exist in honey bees, how many have not been detected in honey bees but are conserved in other species, and how many are entirely novel lncRNAs.
- I would consider representing the findings from Figure 5 in a more visually appealing manner – it is completely impossible to decipher what these networks are telling us, they are illegible.
- Regarding the claims of some lncRNAs being antisense, these claims cannot be made unless the authors have specifically performed strand-specific transcriptomics (see PMID: 28464849).
- The names of each group are very confusing – while reading the manuscript I was unable to figure out what the code for each group meant. I recommend using names for the groups that are meaningful rather than abbreviations that are easy to forget the meaning of. Actually, now that I have read the whole manuscript, I don’t think what each group was is ever mentioned in the entire paper. A description of what each group is will be required. This should be right up front at the beginning of the results section when the authors first start mentioning the group abbreviations.
- The manuscript is lacking a concluding paragraph to sum up all the findings. I see it now, it comes after the methods. This is confusing – if the journal allows it I suggest having the conclusions follow the discussion instead of having the methods in between.
- I struggle with the authors’ claims that they have identified cis-acting and trans-acting regulatory roles, or that some lncRNAs interact with other molecules, etc. No in vitro or in vivo functional analyses have been conducted at all in this study. The authors need to be very careful not to make claims of function or molecular interaction.
- In the field of transcriptomics, it is the gold standard to use two different programs for detection of differential expression. The authors only used edgeR – it is recommended to use edgeR and DESeq2 and then take the consensus (e.g., those that are determined to be differentially expressed by BOTH programs).
- I didn’t feel that the discussion section did a great job of explaining what all of these potential regulatory interactions actually mean to the host-pathogen interaction. The discussion was very long and mostly just listed things like (X lncRNA may interact with Y mRNA, thus it may regulate the expression of Y gene). But what does this regulation mean in context of host-pathogen interactions? The authors need to do a better job of explaining the meaning of these interactions rather than just stating that they may exist. So what?
Minor concerns:
- Inconsistent use of “bp” and “nt” – pick one and be consistent.
- Line 109: “unclosing” is not an appropriate word here. I suggest “identifying” or “examining”.
- Lines 114-115: The authors should introduce what each group is (e.g., what is the difference between AcT1 and AcCK2 etc.?) here because this has not been explained yet. There should be some kind of introduction describing (very briefly) the methods used (e.g., We sequenced the transcriptomes of honey bees infected with N. ceranae [describe the groups here] to identify lncRNAs. There should also be some mention of the time points studied.
- Any figures using red and green to contrast one another should be changed to red and blue to accommodate color blind readers.
- The figure panels should be called out in order. For example, Figure 2A is called out first and then Figure 2E is called out next – what about 2B, C, and D? If the ordering of the figures does not make sense for the order the authors want to discuss the results then the figure panels need to be reordered.
- Line 146: “Venn analysis” is awkward, it is not really an analysis it is a visual representation.
- Line 155: typo “valcano”
- The authors should describe what “cis-acting effect analysis” is on line 60 before the authors introduce the results.
- Make sure to define all acronyms and abbreviations at first use. For example “GO” and “KEGG” need to be spelled out at first use (lines 162 and 170 respectively). These are just examples – please check all abbreviations throughout the manuscript.
- Line 190: typo “mRANs”
- Figure 9 should mention what gene was used for normalization. This can be incorporated into the y-axis: “relative lncRNA levels relative to the levels of X mRNA”.
- Line 329: unnecessary line break
- Line 365: typo “participat”
- Figure 10: Just confirming that the authors have a Biorender license that permits them to publish this honey bee image. A specific academic license is required to use Biorender graphics in publications.
Author Response
Major concerns:
- The lncRNAs identified were said to range in size from 188 bp to 441,800 bp. This does not make sense because lncRNAs, by definition, are larger than 200 bp in length.
Response: Thanks for your helpful comment. As you said, it’s commonly defined that lncRNAs are larger than 200 nt in length. After checking the detailed information about known lncRNAs and novel lncRNAs identified in this work, we detected that the lengths of all novel lncRNAs were larger than 200 bp, but the lengths of a few known lncRNAs were smaller than 200 bp. We inferred that those lncRNAs (smaller than 200 bp in length) conserved in other species was identified following a less strict threshold, with the development of research on lncRNAs, our understanding of lncRNAs became deeper. In addition, it’s currently hard to exclude the possibility that those lncRNAs smaller than 200 bp in length are not lncRNAs. Table S1 was added into the 2.1 section in the revised manuscript to show detailed information of all of the identified lncRNAs.
- This manuscript requires professional English language editing prior to its potential acceptance.
Response: Following your kind comment, we tried our best to improve the language of the manuscript, and ask a native English speaker to help to further polish our manuscript.
- There should be a venn diagram in Figure 1 showing how many of the identified lncRNAs were already known to exist in honey bees, how many have not been detected in honey bees but are conserved in other species, and how many are entirely novel lncRNAs.
Response: According to your valuable recommendation, we downloaded lncRNAs annotated in the referencee genomes of Homo sapiens and Drosophila, which were compared with our previously identified lncRNAs in Apis mellifera ligustica as well as those lncRNAs identified in Apis cerana in this study. Since several days are needed for the process of the comparison of lncRNAs in these species mentioned above, and the deadline of manuscript revision is arriving, we uploaded the revised manuscript and responses in the online system in advance. When we gain the comparison result, we will add the Venn diagram in Figure 1 in the revised manuscript. Thanks.
- I would consider representing the findings from Figure 5 in a more visually appealing manner – it is completely impossible to decipher what these networks are telling us, they are illegible.
Response: Thanks for your helpful suggestion, based on which we replaced Figure 5 with a supplementary table to give more detailed information.
- Regarding the claims of some lncRNAs being antisense, these claims cannot be made unless the authors have specifically performed strand-specific transcriptomics (see PMID: 28464849).
Response: Following your helpful comment, we seriously read this article and deleted contents associated with antisense lncRNAs in the revised version of manuscript.
- The names of each group are very confusing – while reading the manuscript I was unable to figure out what the code for each group meant. I recommend using names for the groups that are meaningful rather than abbreviations that are easy to forget the meaning of. Actually, now that I have read the whole manuscript, I don’t think what each group was is ever mentioned in the entire paper. A description of what each group is will be required. This should be right up front at the beginning of the results section when the authors first start mentioning the group abbreviations.
Response: Thanks for your valuable recommendation, following which we carefully checked the whole manuscript and modified the names of each group.
- The manuscript is lacking a concluding paragraph to sum up all the findings. I see it now, it comes after the methods. This is confusing – if the journal allows it I suggest having the conclusions follow the discussion instead of having the methods in between.
Response: According to your kind comment, the conclusion section was transferred to follow the discussion in the revised manuscript.
- I struggle with the authors’ claims that they have identified cis-acting and trans-acting regulatory roles, or that some lncRNAs interact with other molecules, etc. No in vitro or in vivo functional analyses have been conducted at all in this study. The authors need to be very careful not to make claims of function or molecular interaction.
Response: In this work, expression pattern and putative function of lncRNAs were analyzed to investigate the lncRNA-regulated response of Asian honey bee workers to N. ceranae infestation, mainly using bioinformatic approaches. We believe that our data could provide insights into interaction between Asian honey bee and N. ceranae. Also, based on the findings from this work, several key DElncRNAs identified here, such as TCONS_00022015, TCONS_00032996, and TCONS_00035424, could be used as putative candidates for further functional investigation in the near future. We seriously examined the whole manuscript and made necessary modifications for related descriptions. Thanks.
- In the field of transcriptomics, it is the gold standard to use two different programs for detection of differential expression. The authors only used edgeR – it is recommended to use edgeR and DESeq2 and then take the consensus (e.g., those that are determined to be differentially expressed by BOTH programs).
Response: As you said, to detect differential expression of transcripts is a common method used in some transcriptomic studies. After widely checking relevant documentations, we also found that edgeR was solely utilized in some transcriptomics-associated studies on animals, plants, and microorganisms [Ayala-Ortiz, C. O.; Farriester, J. W.; Pratt, C. J.; Goldkamp, A. K.; Matts, J.; Hoback, W. W.; Gustafson, J. E.; Hagen, D. E. Effect of food source availability in the salivary gland transcriptome of the unique burying beetle Nicrophorus pustulatus (Coleoptera: Silphidae). PLoS One. 2021, 16, e0255660; Ortiz, J. P. A.; Leblanc, O.; Rohr, C.; Grisolia, M.; Siena, L. A.; Podio, M.; Colono, C.; Azzaro, C.; Pessino, S. C. Small RNA-seq reveals novel regulatory components for apomixis in Paspalum notatum. BMC Genomics. 2019, 20, 87; Tomescu, M. S.; Sooklal, S. A.; Ntsowe, T.; Naicker, P.; Darnhofer, B.; Archer, R.; Stoychev, S.; Swanevelder, D.; Birner-Grünberger, R.; Rumbold, K. Transcriptome and proteome of the corm, leaf and flower of Hypoxis hemerocallidea (African potato). PLoS One. 2021, 16, 0253741; Tomescu, M. S.; Sooklal, S. A.; Ntsowe, T.; Naicker, P.; Darnhofer, B.; Archer, R.; Stoychev, S.; Swanevelder, D.; Birner-Grünberger, R.; Rumbold, K. Transcriptome and proteome of the corm, leaf and flower of Hypoxis hemerocallidea (African potato). PLoS One. 2021, 16, 0253741]. In addition, to verify differential expression of lncRNAs identified in this work, 12 randomly selected DElncRNAs were subjected to RT-qPCR detection, the results indicated that the differential expression of these DElncRNAs were consistent with those in the sequencing data. Taken together, it’s believed that detection of differential expression of lncRNAs using edgeR is feasible and reliable.
- I didn’t feel that the discussion section did a great job of explaining what all of these potential regulatory interactions actually mean to the host-pathogen interaction. The discussion was very long and mostly just listed things like (X lncRNA may interact with Y mRNA, thus it may regulate the expression of Y gene). But what does this regulation mean in context of host-pathogen interactions? The authors need to do a better job of explaining the meaning of these interactions rather than just stating that they may exist. So what?
Response: Thanks for your valuable comments, following which we seriously checked many articles relative to honey bee, Nosema, and lncRNA. Accordingly, we seriously rewritten and refined the discussion section combined with the background of bee nosemosis, please see the revised manuscript.
Minor concerns:
- Inconsistent use of “bp” and “nt” – pick one and be consistent
Response: According to your kind comment, we carefully made corresponding corrections throughout the manuscript.
- Line 109: “unclosing” is not an appropriate word here. I suggest “identifying” or “examining”
Response: It was replaced by “examining” in the revised manuscript.
- Lines 114-115: The authors should introduce what each group is (e.g., what is the difference between AcT1 and AcCK2 etc.?) here because this has not been explained yet. There should be some kind of introduction describing (very briefly) the methods used (e.g., We sequenced the transcriptomes of honey bees infected with ceranae[describe the groups here] to identify lncRNAs. There should also be some mention of the time points studied
Response: According to your helpful suggestion, we added the introduction of each group, the used methods, and the mention of time points in the revised version of manuscript.
- Any figures using red and green to contrast one another should be changed to red and blue to accommodate color blind readers
Response: On basis of your kind comments, we carefully modified colors of each figure.
- The figure panels should be called out in order. For example, Figure 2A is called out first and then Figure 2E is called out next – what about 2B, C, and D? If the ordering of the figures does not make sense for the order the authors want to discuss the results then the figure panels need to be reordered.
Response: Corresponding changes were made to make sure that the figure panels were called out in order. Thanks.
- Line 146: “Venn analysis” is awkward, it is not really an analysis it is a visual representation.
Response: The description here was modified according to your helpful comment.
- Line 155: typo “valcano”.
Response: This was corrected here. Thanks.
- The authors should describe what “cis-actingeffect analysis” is on line 60 before the authors introduce the results.
Response: Following your kind comment, we added the explanation of “cis-acting effect analysis” in the introduction section in the revised manuscript.
- Make sure to define all acronyms and abbreviations at first use. For example “GO” and “KEGG” need to be spelled out at first use (lines 162 and 170 respectively). These are just examples – please check all abbreviations throughout the manuscript.
Response: Following your helpful comment, we carefully examined the whole manuscript and defined all acronyms and abbreviations at first use in the revised version.
- Line 190: typo “mRANs”.
Response: This was corrected here.
- Figure 9 should mention what gene was used for normalization. This can be incorporated into the y-axis: “relative lncRNA levels relative to the levels of X mRNA”.
Response: Corresponding mention was added into the title of y-axis in Figure 9. Thanks.
- Line 329: unnecessary line break.
Response: The line break was deleted.
- Line 365: typo “participat”.
Response: This word was corrected in the revised manuscript.
- Figure 10: Just confirming that the authors have a Biorender license that permits them to publish this honey bee image. A specific academic license is required to use Biorender graphics in publications.
Response: Thanks for your kind reminder. We replaced the honey bee image by drawing a new one using AI software.

Reviewer 2 Report
No Comments for Authors
Author Response
Thanks.
Reviewer 3 Report
The original manuscript by Wang et al., investigate the role of lncRNAs in microsporidian infestation in honeybees. The topic is poorly studied, but it is of a great importance. The research is very detailed and includes large amount of information. To be consider for publication, the authors must answer several questions.
Have the RNA-seq data been deposited in a public domain or a database has been generated?
Can authors provide the agarose gels of the PCR products to prove the specificity of the primers used for RT-qPCR experiments?
Can the authors provide RT-qPCR validation for DEmRNAs as well?
The SD bars in Fig. 9 are missing.
It will be interesting if the authors mention the economic importance of microsporidian infection in bees.
Author Response
Response to Reviewer 3:
- Have the RNA-seq data been deposited in a public domain or a database has been generated?
Response: Thanks for your kind comment. In fact, raw data generated from RNA-seq had been deposited in the NCBI SRA database and linked to the BioProject number: PRJNA562787, which was mentioned in the 4.2 section (line 505-506).
- Can authors provide the agarose gels of the PCR products to prove the specificity of the primers used for RT-qPCR experiments?
Response: Following your requirement, we added the agarose gel electrophoresis result (Figure S2) in the revised manuscript (see below). The result verified the specificity of the primers used for RT-qPCR experiments. Thanks.
- Can the authors provide RT-qPCR validation for DEmRNAs as well?
Response: In a previous work, we have already performed RT-qPCR detection of a total of 16 DEmRNAs, the results were in accordance those in the transcriptome data, verifying the reliability of sequencing data [Xing, W.; Zhou, D.; Long, Q.; Sun, M.; Guo, R.; Wang, L. Immune Response of Eastern Honeybee Worker to Nosema ceranae Infection Revealed by Transcriptomic Investigation. Insects. 2021, 12, 728.]. The same transcriptome data was used in this work.
- The SD bars in Fig. 9 are missing.
Response: Thanks for your kind comment. We have added SD bars and figure notes in Figure 9.
- It will be interesting if the authors mention the economic importance of microsporidian infection in bees.
Response: Based on your valuable comment, we added corresponding descriptions in the introduction section in the revised version of manuscript.

Round 2
Reviewer 3 Report
-